# The impact of digital communication and data exchange on primary health service delivery in a small island developing state setting

Kaye Borgelt[1], Taniela Kepa Siose[2‡], Isaia V. Taape[2‡], Michael Nunan[3‡], Kristen Beek[4‡], Adam T. Craig[4] *

**1** Digital health consultant, Melbourne, Victoria, Australia, **2** Ministry of Health Social Welfare and Gender Affairs, Tuvalu, **3** Beyond Essential Systems, Melbourne, Victoria, Australia, **4** School of Population Health, University of New South Wales, New South Wales, Australia

☯ These authors contributed equally to this work.
‡ TKS, IVT, MN, and KB also contributed equally to this work.
* adam.craig@unsw.edu.au

**Data Availability Statement:** All data are in the manuscript and/or supporting information files.

**Funding:** This work was supported by a grant to ATC from the Asia Pacific Observatory for Health

## Abstract

Tuvalu is one of the smallest and most remote countries in the world. Due partly to its geography, the limited availability of human resources for health, infrastructure weaknesses, and the economic situation, Tuvalu faces many health systems challenges to delivering primary health care and achieving universal health coverage. Advancements in information communication technology are anticipated to change the face of health care delivery, including in developing settings. In 2020 Tuvalu commenced installation of Very Small Aperture Terminals (VSAT) at health facilities on remote outer islands to allow the digital exchange of data and information between facilities and healthcare workers. We documented the impact that the installation of VSAT has had on supporting health workers in remote locations, clinical decision-making, and delivering primary health more broadly. We found that installation of VSAT in Tuvalu has enabled regular peer-to-peer communication across facilities; supported remote clinical decision-making and reduced the number of domestic and overseas medical referrals required; and supported formal and informal staff supervision, education, and development. We also found that VSAT's stability is dependent on access to services (such as a reliable electricity supply) for which responsibility sits outside of the health sector. We stress that digital health is not a panacea for all health service delivery challenges and should be seen as a tool (not the solution) to support health service improvement. Our research provides evidence of the impact digital connectivity offers primary health care and universal health coverage efforts in developing settings. It provides insights into factors that enable and inhibit sustainable adoption of new health technologies in low- and middle-income countries.

## Author summary

While functional health information systems are central to effective and efficient service delivery, many low- and middle-income countries lack the information communication

Systems and Policies (Grant reference: 2020/1002187-0). The funders had no role in study design, data collection and analysis, decision to publish, or preparation of the manuscript.

**Competing interests:** The authors have declared that no competing interests exist.

technology infrastructure required for the digital collection and exchange of health data. This issue is particularly relevant to Small Island Developing States where geography, small population sizes, reliance on extra-national expertise, and supply-chain limitations are barriers to digital health adoption. This paper explores the impact installation of very small aperture terminal (VSAT) technology at health facilities across the remote South Pacific Island nation of Tuvalu has had on the delivery of primary health care and on efforts to achieve universal health coverage. We found that VSAT has enabled regular peer-to-peer communication between health workers on outlying islands and the county's only hospital enhancing clinical decision-making and reducing the number of domestic and overseas medical referrals required. It has also led to increased formal and informal staff supervision and training opportunities and aided networking with colleagues overseas. We found that successful adoption of new technology requires systems to be reliable and convenient for end-users. We suggest simple solutions harnessing ubiquitous technology (such as smartphones coupled with commercial communication apps, such as Zoom) may be more readily adopted by healthcare workers than dedicated platforms. While the introduction of VSAT in Tuvalu has overcome some long-standing challenges, stakeholders stressed that technology is not a panacea for all the problems they face.

## Introduction

Tuvalu, located in the west-central Pacific Ocean halfway between Australia and Hawaii (Fig 1), is one of the smallest and most remote countries in the world. Tuvalu has a total landmass of 26 square kilometres spread over nine low-lying coral atolls and a population of just 10645 [1]; over half (63.1%) of which live on the main island of Funafuti [1]. With the highest point only five meters above sea level, Tuvalu and its population are considered highly vulnerable to the impacts of climate change [2–4].

With an average annual income of ~USD4330 per person, Tuvalu is classified as a low-middle-income country by the World Bank [5]. Life expectancy at birth was 68.1 years, and infant mortality was 36.6 deaths per 1,000 live births in 2018 [6]. Tuvalu has a true universal health care system where Tuvaluans have access to free health care through a network of facilities located across the atolls. The national health system comprises one hospital (Princess Margaret Hospital [PMH]) on Funafuti (the capital island) and one primary health clinic on each of the eight outer islands. Princess Margaret Hospital is a fifty-bed facility providing primary and secondary level care and limited diagnostic services. In 2021, Tuvalu's medical-trained workforce comprised 13 general doctors and three specialists (an Anesthetist, Obstetrician and Gynecologist, and a General Surgeon). Medically trained staff are not routinely posted to primary care clinics on outer islands; Registered Nurses and Nursing Assistants staff these facilities. The Ministry of Health covers patients' international medical transfer and treatment costs if required.

In 2018, Government expenditure on health was ~USD8.0M, 15.8% of the total government budget; ~USD4.7M, or 58.8%, of this was spent on overseas medical transfers and treatment [6]. The most common destinations for overseas referrals were Fiji, India, and Malaysia [6].

An advancement in information communication technology (ICT) that has gained popularity in recent years and is anticipated to change the face of health care is digital health. Digital health is a broad term used to describe the use of ICT, in all its forms, to support and enhance the provision of health service delivery. The unanimous approval by WHO Member States of the 'Resolution on Digital Health' [7] at the 2018 World Health Assembly and the Regional Committee for the Western Pacific's endorsement of the Regional Action Agenda on

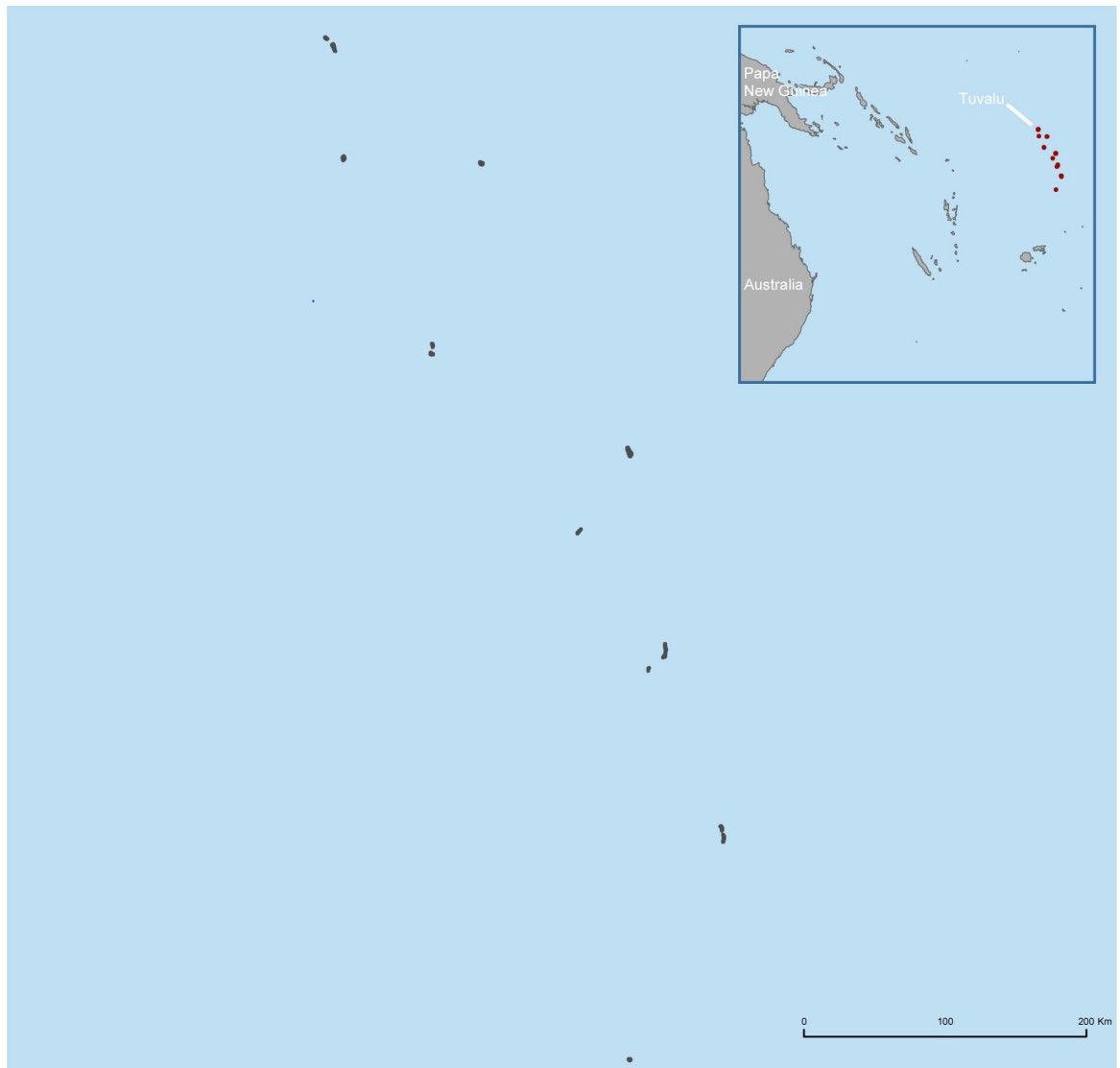

**Fig 1. Map of Tuvalu and Tuvalu's location in the South Pacific (insert).** Caption: Source: Center for International Earth Science Information Network (CIESIN), Columbia University, https://commons.wikimedia.org/wiki/File:Tuvalu_Input_Administrative_Boundaries_(5457156727).jpg, CC BY 2.5 [http://creativecommons.org/licenses/by/2.5/].

Harnessing e-Health for Improved Health Service Delivery [8] demonstrates a recognition of the value digital technology offers health service development and the commitment of decision-makers to exploring its use. There are many examples of the use of ICT to support health care delivery in low- and middle-income settings including for delivery of telemedicine [9–11], targeted patient communication [12–15], supply chain management [16,17], workforce training [18–20], clinical decision-making support, [21,22] data collection and management [16,18,19,23,24], and healthcare workers-to-healthcare worker communication and data exchange [25–27] including during disaster situations [28].

Tuvalu's first digital health strategy was drafted in 2019 [29]. As part of this strategy, plans to install Very Small Aperture Terminals (VSAT) at PMH and primary health clinics on the eight outer islands were made. VSAT are two-way ground stations with a dish antenna used to transmit and receive digitized data over a satellite communication network (Fig 2) [27,30,31].

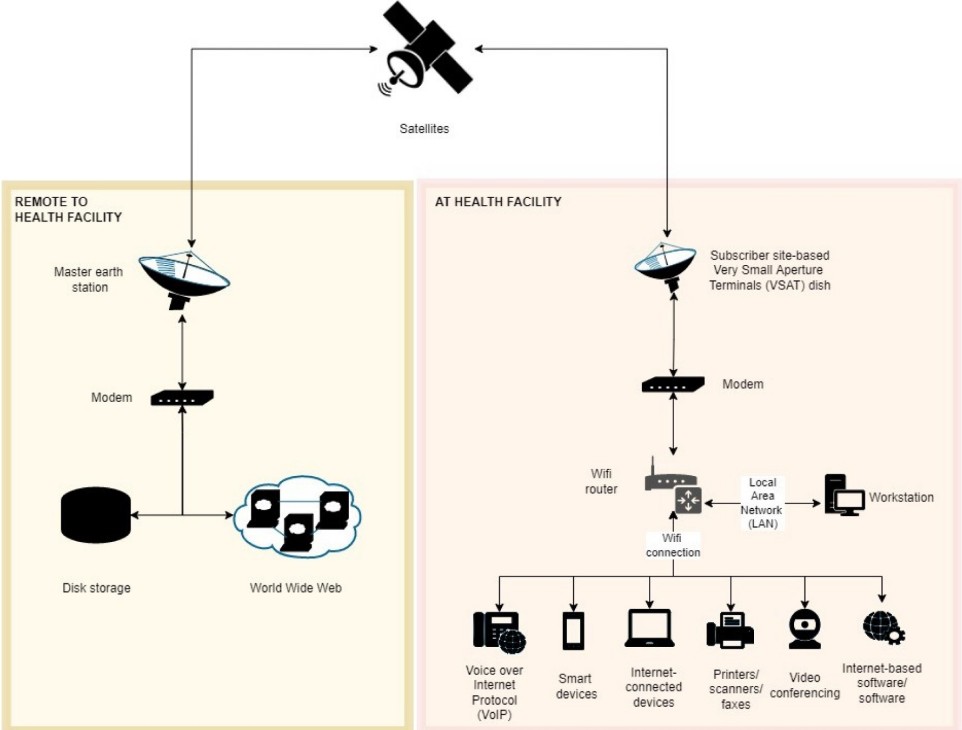

**Fig 2. Schematic of the Tuvalu Very Small Aperture Terminal (VSAT) system.** Caption: This figure is an author elaboration based on Fig 7 in [27]. Figure created with diagrams.net.

Compared to other modes of data transmission–such as fiber-optics–satellite communications are slow and can experience long transmission delays. In Tuvalu, it is envisaged that VSAT will provide foundational infrastructure on which digital health solutions designed to address various health systems and service delivery challenges may be implemented. These challenges include (but are not limited to) the direct delivery of patient care, health worker-to-health worker consultation, patient referral and record exchange, staff training, and stock and supply chain management.

This paper presents research exploring the experiences of healthcare workers with the introduction of VSAT in Tuvalu, and the impact digital connectivity has had on the delivery of primary health care and efforts to achieve universal health coverage.

## Results

Eight key informants were interviewed, three one-to-one and five in small groups. Interviewees included all nursing staff that worked at the three (of the eight) outer island primary health clinics at which VSAT has been installed, clinical staff from PMH, and managerial staff of the Ministry of Health responsible for the development and implementation of VSAT and the national digital health strategy. We estimate that 80% of active VSAT stakeholder were interviewed. Seven informants were female.

Five main themes emerged. These are discussed below.

### VSAT enabled peer-to-peer communication between and across facilities

A commonly expressed theme and perhaps the most transformative impact the installation of VSAT has had in Tuvalu is the capacity and relative ease with which health workers on outer

islands have been able to in real or near real-time consult with medical staff at PMH. Previously, the only means of direct communication was via a single incoming and outgoing landline telephone line. Connectivity by this means was dependent on functioning telecommunications infrastructure at both ends of a call at the same time, the availability of hospital staff to answer incoming calls at all hours, and the on-call doctor being on hand to respond. These seemingly minor issues were reported to be persistent and significant barriers to reliable real-time communication, reducing the willingness of health workers on outer islands to call for assistance as the means was perceived as cumbersome and often did not deliver the advice required on time.

Interviewees' report that VSAT-enabled phone and email communication are more reliable and, as they allow a direct line of communication with a clinician at PMH, tend to result in faster, more considered, and clinically more helpful advice. This sentiment was captured well by a doctor who said, "[due to VSAT] everyone has access to the internet and the communication has become easy and quick, and the consultants (i.e., specialist doctors at PMH) are prepared to discuss a case directly with the nurse on the island. [This is] not like before where it (sic) had to come through the on-call doctor . . . the response is way easier, and the management of patients has improved a lot." (Participant No. 8).

Interviewees reported most often accessing the VSAT network from a personal smart device (typically a smartphone) and thereby having constant access to email, text and/or video-conferencing applications while on a health campus. The immediacy of access to a device was noted as an enabling factor as it made use of the technology both familiar and convenient and–through regular use–has built staff members' capacity and willingness to embed VSAT-enabled communication into routine practice. A doctor interviewed commented, "[rather] than trying to get a call from the outer islands now we [doctors] just check our phones and the first one that sees a message from the outer islands requesting our help will just reply. . .It is much better than before where we had to run all the way to where the radio is" (Participant No. 7).

Post-installation upgrades to the VSAT system have enabled faster data transfer (upload/download speeds are now ~2 Megabytes per second, an improvement on ~250 kilobytes per second previously experienced) broadening the range of media that could be transmitted. This has seen larger files (such as photo and video files and more stable videoconferencing) shared between facilities and led to richer intelligence sharing to inform remote consultation and differential diagnosis. A doctor commented, "the internet is a lot faster now and nurses can take [and send] pictures and videos so it is a lot easier for us [doctors] to diagnose and make a clinical decision" (Participant No. 8).

## VSAT impact on the number of domestic and overseas medical referrals required

Interviewees reported that enhanced capacity to consult with clinicians at PMH due to the introduction of VSAT had resulted in fewer domestic and overseas medial referrals, saving patients and staff considerable time and the health system money. One doctor said, "The practice before [the installation of VSAT] was that the nurse would just send a patient [to PMH] if she thought they needed to be seen by a doctor. Now we have a procedure in place that says that remote consultation or case discussion [between nurses at outer island primary health clinic and doctor at PMH] needs to happen before any patient is transferred. This has resulted in many more patients receiving their treatment on the [outer] islands, or, if a patient does need to come to PMH, we know about them in advance. This helps with [the delivery of] care" (Participant No. 7). Another said, "[VSAT] has been really helpful and reduced referrals from

the outer islands because treatment can be given [remotely]. The doctors [at PMH] can see the patient on the outer island and diagnose and treat them from there. This contributes to less referrals from outer islands and less disruption for the patient" (Participant No. 2).

Beyond initial referral, use of VSAT has led to changes in the way routine clinical reviews of patients located on outer islands are managed. Rather than patients being required to attend PMH on Funafuti for their clinical review, VSAT-assisted telemedicine and/or peer-to-peer consultation has, in some instances, been used to provide community-based service provision.

Along with the domestic setting, VSAT-enabled communication has been used to support decision-making regarding the need for overseas medical transfers. Treating doctors at PMH have used VSAT to communicate directly with overseas-based specialists and, on occasions, perform remote telehealth consultations. A PMH doctor said, "Earlier this year, we started discussing cases with New Zealand consultants. We were able to set up a zoom call between the patient and the consultant as well as discussing the case, doctor-to-doctor. The opportunity to discuss complex cases with overseas specialists has streamlined the referral process by allowing the receiving doctor to be involved in clinical decision-making for patient's case plan, including the need for medical transfer" (Participant No. 8).

## VSAT supported formal and informal staff education and development

In addition to direct peer-to-peer consultations, VSAT-enabled communication was reported to have facilitated incidental mentorship and learning opportunities. One administrator said, "the clinical side of nursing is improving because they can see the doctor at PMH and discuss the case with them. I think the doctor gets a better understanding of the issue through the eyes of the nurse and, together, come up with a treatment plan" (Participant No. 3).

In addition, VSAT has increased access to formal workforce development opportunities, particularly for staff located on the outer islands. An interviewee reported "Earlier this year, we started to deliver weekly continuing medical education (CME) by Zoom. One of us [doctors] would present a topic to the outer island nurses, then we'd discuss it. I found that this an efficient way to support their learning" (Participant No.8). Another PMH-based interviewee noted that "internet access at health facilities had facilitated international collaboration for staff development, citing that weekly CME session with staff from a hospital in Taiwan as an example" (Participant No. 1).

Challenges were also noted. A lack of familiarity and fundamental information technology skills among some was identified as a barrier to accessing and utilizing the system. This was a particular issue for nurses on the outer islands who reported being less experienced or comfortable with using technology than their Funafuti-based colleagues. One of the participants commented, "one of the big challenges, when we started off, was the nurses on the islands did not know how to use a computer; they were only used to using a paper and pen. So, I gave the nurses some lessons, you know just some simple things like how to underline or use caps locks. When we started out, it was quite challenging for a nurse to sign into the zoom or use a passcode, sometimes they just panicked. So, I tell them it is just like going in to Facebook and put in your username" (Participant No. 1). Recognizing this challenge, the Ministry of Health engaged an information communication support technician on each of the islands to assist staff in using the new technology.

## VSAT remains vulnerable to broader system disruptions

A pertinent point raised was that VSAT's stability is vulnerable to a range of internal and external technical and operational issues, and hence solely investing in the equipment is inadequate. An example provided by several interviewees was the requirement for a consistent electricity

supply, something that due to fuel shortages (to run electricity-producing generators) and equipment breakdowns is not guaranteed on outer islands. Another respondent noted that the humid tropical and coastal environment of Tuvalu accelerates the deterioration of ICT infrastructure and means that maintenance is required more frequently, which in the absence of stock supply and staff expertise is a challenge. One interviewee commented, "Here in Tuvalu, we live very close to the sea, and the air is really salty. You can see it here that all the computers that are not in an air-conditioned room quickly rust and have stains within a few months" (Participant No. 1). The impact of climatic factors–in particular, rain–on VSAT function are discussed in the literature [32].

### VSAT is a tool and not a solution to health information exchange needs

As highlighted above, the introduction of VSAT in Tuvalu has provided an important means for peer-to-peer communication and ad-hoc information exchange. While providing opportunity, the system-wide networking and integration of health information systems has not yet occurred. That is, patient medical records-, service delivery-, stock and supply- and other records remain facility-based and, for the most part, paper-based. While PMH uses an electronic system for medical record management, it operates through a closed internal network that is not available to other sites. While plans are in place to create a tailored and integrated national health information system (HIS) that will allow simultaneous multi-user and multi-site access–and thereby streamline whole-of-health system data collection, storage, and accessibility–a lack of domestic ICT and database design expertise and COVID-19-related global disruptions have seen these plans delayed.

Attention was drawn to the recent adoption of mSupply (www.msupply.org.nz), a licensed medical inventory software that supports drug and stock procurement and tracking logistics, in Tuvalu with the lessons learned from its implementation expected to be informative for broader HIS roll-out across the country.

## Discussion

Our research involved interviews with eight stakeholders (representing an estimated 80% of users) of the VSAT system in Tuvalu to understand the impact its introduction has had on health system function in the country. Our analysis identified five main themes, that VSAT has: enabled regular peer-to-peer communication across facilities; supported remote clinical decision-making and reduction in the number of domestic and overseas medical referrals required; and supported formal and informal staff education and development. We also found that VSAT's stability is dependent on access to services (such as a reliable electricity supply) for which responsibility sits outside of the health sector.

An inability to provide adequate professional supervision and clinical decision-making support to health staff working in rural and remote parts of the world is a significant challenge to equitable health care delivery and the achievement of UHC. Our research found that the introduction of VSAT in Tuvalu has profoundly impacted the frequency and nature of peer engagement and is facilitating formal and informal opportunities for clinical and professional development. The capacity to communicate has, according to respondents, led to improved health service efficiency and the quality and continuum of care that can be provided. Other countries facing similar challenges, including establishing reliable communication links with staff in rural and remote areas, and providing clinical decision-making, supervision, and professional development support may learn from the Tuvalu experience.

VSAT, while considered a suitable technology for Tuvalu, where land-based digital telecommunications infrastructure is lacking, is one of many ICT options available. Where land-

based and mobile digital networks are present, authorities may find it more cost-effective to invest in the facility-based infrastructure and personal internet connectable devices required to access commercial telecommunications services. Communication technology is rapidly evolving, and with the launch of new-generation high-capacity satellites costs associated with data transfer will likely reduce. Assessment of the most appropriate mix of ICT for a given context is required with careful consideration given to both upfront and ongoing cost and maintenance requirements.

Regardless of ICT infrastructure adopted, authorities must ensure reliability, convenience, and end-user experience are considered in the design to maximize uptake. Interviewees noted that access to and familiarity with digital tools was an enabling factor to their adoption. This suggests that simple solutions harnessing widely used technology (e.g., smartphones coupled with commercial communication apps such as Zoom, Facebook Messenger, or WhatsApp) may be more readily adopted tools for digital communication than dedicated customized platforms. While this raises data privacy and security concerns, decision-makers need to acknowledge the advantages of leveraging ubiquitous technology, weigh the pros and cons, and–where required–establish policy and protocol to underpin the appropriate exchange of health data.

While the introduction of VSAT in Tuvalu has overcome some long-standing challenges to staff communication, stakeholders interviewed stressed that the technology is not a panacea for all the problems they face. They suggest that VSAT technology be seen as a tool (and not the solution) to address the broader system challenges related to communication and data exchange. They advise that investments in a complete suite of interconnected and interoperable digital solutions, including digital medical record management, telemedicine, and ePharmacy and eLaboratory tools is required for full effect. Our research highlights that technology-based interventions rely on the stability of other essential services, such as electricity, for their operation. It warns that while, digital approaches may make sense in theory, in resource-constrained settings (such as in small island developing states) there are often multiple and complex development challenges to overcome to ensure sustainable and scalable adoption of digital health. The impact of such challenges should not be underestimated and require digital health stewards to take a multisector whole-of-systems approach to find solutions.

Tuvalu is at the early stages of digital health adoption. While VSAT has made an important contribution, its full capacity has not yet been realized. Capitalizing on the opportunity digital data exchange offers by integrating other systems such as an electronic Logistics Management Systems for commodity and drug supply chain management, direct patient-provider video-conferencing facilities for remote diagnosis and delivery of care, and a Health Management Information System for the collection, storage, aggregation, interpretation, and presentation of routine and ad hoc health indicators is advised as a next step. The adoption of digital tools to support the delivery of health services needs and be done with full consciousness of the financial and human resources required. In settings where the human resources or funds required for digital health are limited, health authorities may need to partner with development or philanthropic organizations to ensure capacity is available for scalable and sustainable adoption. Integration of digital tools will likely disrupt established (and potentially) fragile health service processes. Designers must recognize the destabilizing impact change may have on these processes and should set up mechanisms to maintain essential services (such as supply chain management and medical record-keeping) during the transition period.

With the rapid development of telehealth-based service provision globally, it is worth investigating how telemedicine services may be used to complement (or supplement) health care delivery in Tuvalu and other developing settings that lack human resources for health. One model that may be worth exploring is reliable digital access to a formalized network of expert consultation through which healthcare workers may seek professional advice where required.

Another model that may work is periodic teleconferencing with specialists for peer clinical review and treatment planning. Further, as identified by our research, telehealth-based consultation between sending and receiving healthcare workers before transfer allows for complete handover, collective decision making about a patient's course of treatment, and an enhanced continuum of care.

Our research shows that Tuvalu (and we expect other small island developing states in the Pacific and elsewhere) lacks the administrative instruments (the legal, policy and procedures) to govern the collection, storage, and use of digitalized health data. These instruments provide the backbone for good health data stewardship and must be priorities. While each Pacific country is unique and requires tailored laws and regulations, there may be an opportunity to work collaboratively to develop principle-based digital health governance resources, including flexible templates for countries to work from. This approach, if aligned with global and regional frameworks for digital health development (such as the *WHO Global Strategy on Digital Health*, *2020–2025*, [33] the *Regional Action Agenda on Harnessing E-Health for Improved Health Service Delivery in the Western Pacific*, [8]) could build on the well-established mechanisms for regional collaboration that operate in the Pacific.

While insightful, we were not able to verify our findings due to the lack of a monitoring and evaluation associated with VSAT implementation, or the national digital health strategy more broadly. To ensure that VSAT is meeting its goals and is being delivered in an efficient and effective way, the establishment of an indicator-based monitoring and evaluation framework is warranted. Using both qualitative and quantitative methods, such a framework may collect and analyze data on aspects of performance relevant to the Tuvalu Government. These may include (but are not limited to) the impact VSAT has on access to and the quality of primary health care, on UHC goals; on workforce support; training, and development; and on health sector expenditure. Information produced may also inform digital health development efforts, globally. Qualification of the reduction in cost associated with overseas referrals alone would help justify anticipated expenditures on VSAT and other digital health initiatives.

This research is not without limitations. First, participants, while representing a significant portion (estimate of ~80%) of all stakeholders, were not randomly selected and hence selection bias may have been introduced. Second, due to logistical constraints, interviews were done online using Zoom, and it was not always possible to conduct interviews one-to-one, potentially influencing interviewee responses. Third, our research is qualitative in nature and, as such, it captured participants' views and experiences; we were not able to verify claims made. Finally, our work focused on the Tuvalu experience as a case study and, while sharing features with other countries, there are likely contextual differences that need to be considered when interpreting results. All this said, our work is the first to explore the use of VSAT as a digital tool to support health service delivery in a small island developing state context and as such is novel. The results and discussion provide insight into important and practical issues others considering a digital enhancement to health systems ought to consider.

Our research provides evidence of the impact digital connectivity offers efforts to improve primary health care and UHC in low resources and complex service delivery settings. It provides insights into factors that enable and inhibit sustainable adoption of digital health for health sector development.

## Materials and methods

### Participants

Interviewees were purposively selected in consultation with the Tuvalu Deputy Secretary, Ministry of Health, Social Welfare and Gender Affairs (co-author T.K.S.). Informants were selected

for their knowledge of the system and to ensure representation from a range of roles, experiences, and perspectives. To be eligible, informants had to be an employee or advisor of the Tuvalu Ministry of Health and be working directly with the VSAT system.

## Data collection

Interviews were conducted one-to-one or in small groups via video conference; KB conducted all interviews. The interviews were semi-structured and took 30- to 65-minutes to complete; all interviews were recorded and transcribed verbatim. Detailed notes were also kept. A semi-structured interview format was chosen to allow participant-initiated discussions to flow and to be captured while also ensuring key data were collected.

An interview tool was developed based on previously published examples (S1 File). The tool elicited information about (i) respondents' demographics; (ii) workplace experience; and (iii) views and examples of the impact the introduction of VSAT has had on work practices. We pilot tested the tool with three people and made minor adjustments based on feedback received.

## Data analysis

Interview data (i.e., the transcripts and detailed notes kept by the interviewer) were thematically analyzed using a general inductive approach [34]. We designed a coding frame that categorized and re-categorized data according to emergent themes. Verbatim quotes were extracted from the transcriptions and used to highlight salient points.

## Ethics

The Tuvalu Ministry of Health and the University of New South Wales Human Health Ethics Committee (Ref: HC210190) provided approval to undertake the research. Participants were provided with written and verbal information about the research. Informed consent was obtained from each participant.

## Supporting information

**S1 File. Key informant interview data collection tool.**
(DOCX)

## Acknowledgments

We thank the health facility staff that generously provided their time and insights.

## Author Contributions

**Conceptualization:** Kaye Borgelt, Adam T. Craig.

**Data curation:** Kaye Borgelt.

**Formal analysis:** Kaye Borgelt, Adam T. Craig.

**Funding acquisition:** Adam T. Craig.

**Investigation:** Kaye Borgelt, Adam T. Craig.

**Methodology:** Kaye Borgelt, Adam T. Craig.

**Project administration:** Adam T. Craig.

**Resources:** Adam T. Craig.

**Supervision:** Adam T. Craig.

**Visualization:** Adam T. Craig.

**Writing – original draft:** Kaye Borgelt, Adam T. Craig.

**Writing – review & editing:** Kaye Borgelt, Taniela Kepa Siose, Isaia V. Taape, Michael Nunan, Kristen Beek, Adam T. Craig.

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
