## [Decision Letter · Decision Letter 0]

1 Jul 2022

PDIG-D-22-00057

The impact of digital communication and data exchange on primary health service delivery in a small island developing state setting

PLOS Digital Health

Dear Dr. Craig,

Thank you for submitting your manuscript to PLOS Digital Health. After careful consideration, we feel that it has merit but does not fully meet PLOS Digital Health's publication criteria as it currently stands. Therefore, we invite you to submit a revised version of the manuscript that addresses the points raised during the review process.

In addition to addressing each reviewers' comments, please make sure the manuscript is in the required PLOS Digital Health format, including the use of figures and formatting of citations.

Please submit your revised manuscript within 30 days . If you will need more time than this to complete your revisions, please reply to this message or contact the journal office at digitalhealth@plos.org. Please include the following items when submitting your revised manuscript:

We look forward to receiving your revised manuscript.

Kind regards,

Heather Mattie

Academic Editor

PLOS Digital Health

Journal Requirements:

2. Please update your online Competing Interests statement. If you have no competing interests to declare, please state: “The authors have declared that no competing interests exist.”

3. Please provide separate figure files in .tif or .eps format and remove any figures embedded in your manuscript file. Please also ensure that all files are under our size limit of 10MB.

For more information about how to convert your figure files please see our guidelines: https://journals.plos.org/digitalhealth/s/figures

4. All figures and supporting information files will be published under the Creative Commons Attribution License (creativecommons.org/licenses/by/4.0/). Authors retain ownership of the copyright for their article and are responsible for third-party content used in the article. 

Figure 1: please (a) provide a direct link to the base layer of the map used and ensure this is also included in the figure legend; (b) provide a link to the terms of use / license information for the base layer. We cannot publish proprietary or copyrighted maps (e.g. Google Maps, Mapquest) and the terms of use for your map base layer must be compatible with our CC-BY 4.0 license. 

Please upload any written confirmation as an 'Other' file type. It must clarify that the copyright holder understands and agrees to the terms of the CC BY 4.0 license; general permission forms that do not specify permission to publish under the CC BY 4.0 will not be accepted. Note that uploading an email confirmation is acceptable.

Additional Editor Comments (if provided):

Reviewers' comments:

Reviewer's Responses to Questions

**Comments to the Author**

1. Does this manuscript meet PLOS Digital Health’s publication criteria? Is the manuscript technically sound, and do the data support the conclusions? The manuscript must describe methodologically and ethically rigorous research with conclusions that are appropriately drawn based on the data presented.

Reviewer #1: Partly

Reviewer #2: Partly

2. Has the statistical analysis been performed appropriately and rigorously?

Reviewer #1: No

Reviewer #2: I don't know

3. Have the authors made all data underlying the findings in their manuscript fully available (please refer to the Data Availability Statement at the start of the manuscript PDF file)?

Reviewer #1: Yes

Reviewer #2: Yes

4. Is the manuscript presented in an intelligible fashion and written in standard English?

Reviewer #1: Yes

Reviewer #2: Yes

5. Review Comments to the Author

Reviewer #1: --REVIEW-- 19.04.2022

Satellites have a wide range of applications, of which the education, medical, banking, oil, and government sectors (such as crisis management, as exemplified by Japan) can be pointed out. They are used to provide communication wherever access to traditional systems is impossible or difficult (mountainous areas, remote regions, island countries such as Tuvalu, etc.). With the help of satellites and VSAT systems, the cheapest global communication system can be applied. For this reason, the use of cheap and small VSAT antennae is so common in such locations, and often becomes the only possibility of communication. In practice, the VSAT systems are widely applied for filling so called “white spots” of access to the Internet or telephones (there are special government programs dedicated for this purpose, e.g. indicated by the Authors of the Tuvalu's digital health strategies). 

Authors write “VSAT, while considered a suitable technology for Tuvalu, where land-based digital telecommunications infrastructure is lacking, is one of many ICT options available. Where land-based and mobile digital networks are present, authorities may find it more cost-effective to invest in the facility-based infrastructure and personal internet connectable devices required to access commercial telecommunications services”.

In recent years, such terminals have been used to access broadband Internet (e.g. from the HTS KA-SAT satellite). The implementation of CONECT VHTS may allow prices to decrease to those offered by fiber Internet providers in the long term. And this should be emphasized by the Authors.

The article presented by the Authors is valuable and in my opinion deserves to be published. As the medical staff of Tuvalu in 2021 included 13 general doctors and three specialists (an anesthesiologist, an obstetrician, and a gynecologist), certainly access to modern VSAT technologies will help to raise the standard of living of the inhabitants of such remote places, including through telemedicine, online consultations, remote surgeries, or access to many Internet-based services. 

However, before publication, it would be useful to consider improving certain elements of the article. I have provided a list of them below.

1). The article has an unusual structure first there is an introduction and then immediately afterwards a results section. It would be worthwhile to consider correcting the content layout.

2). The research methodology (missing element) and limitations of the study should be described beforehand. 

3). The comparison criteria should be better specified.

4). Summary should be at the end of the article.

5). It would be useful to consider adding detailed data on the satellite systems/solutions used in Tuvalu (the data currently presented is residual).

6). The most important comment! It would be helpful to illustrate the data in the form of pie charts (the missing element). As it is, presented descriptions without diagrams are difficult to read. In each of the subsections it would be worthwhile to supplement this. Also, there is a lack of exact percentage statistics in the five main themes considered. This should be corrected.

7). It would be useful to show reference to other reports on ICT use within the area (examples of methodology can be found in many works).

8). Pls complete the bibliography on the use of VSAT systems e.g.:

M. Sasanuma, H. Uchiyama, T. Nagoya, M. Furukawa, T. Motohisa, Research and development

of very small aperture terminals (VSAT) that can be installed by easy operation during disasters – Issues and the solutions for implementing simple and easy installation of VSAT earth station,

IEICE Tech. Rep., Vol. 112 (2013), pp. 1–3.

N. Suematsu, H. Oguma, S. Eguchi, S. Kameda, M. Sasanuma, K. Kuroda, Multi-mode SDR VSAT against big disasters, in: European Microwave Conference, EuMC, Nuremberg, 2013. doi: 10.23919/EuMC.2013.6686788.

S. Kameda, T. Okuguchi, S. Eguchi, N. Suematsu, Development of satellite-terrestrial multimode VSAT using software defined radio technology, in: Asia-Pacific Microwave Conference, IEEE, Sendai, 2014.

9). Authors write “VSAT remains vulnerable to broader system disruptions” – it would be worthwhile to justify it. There are many resources on the Internet. Examples of literature are here: https://doi.org/10.15244/pjoes/73907, https://doi.org/10.15244/pjoes/73906, https://doi.org/10.15244/pjoes/73906. It would be useful to complete the bibliography in this area as well.

10). There are minor editing errors in the article, such as a period in the wrong place after citations.

11). Figure 2 is directly taken from the article cited in the bibliography, i.e. Wilk-Jakubowski J (2021) A review on information systems engineering using vsat networks and their development directions. Yugosl J Oper Rres 31 (3): 409–428. doi:10.2298/YJOR200215015W. The Authors did not provide a source for the figure in the caption. If they have the last publisher's approval it should be sent to the publisher. Pls provide a footnote to this caption, i.e. [12] to avoid plagiarism.

Reviewer #2: You need explain these items.

6. 1. Communication satellites are physical connections and have a large transmission delay compared to submarine cables.

Do you not use video for satellite teleconsulting? In general, for real-time meetings, etc., use broadcast protocols such as UDP-IP will be effective for the communication data speed. 

What kind of communication protocol is it? 

6. 2. The satellite communication with TCP-IP protocol environment, what kind of connection method you are using? for example, DAMA (Demand Assigned Multiple Access) or so on.

6. 3. How effective is the exchange of data between remote islands and central hospital where the movement of patients is extremely small?

In terms of cost, isn't it cheaper to have a patient carry a medium such as a CD-ROM ?

6.4. There are few references, please add more papers.

6. PLOS authors have the option to publish the peer review history of their article (what does this mean?). If published, this will include your full peer review and any attached files.

**Do you want your identity to be public for this peer review?** For information about this choice, including consent withdrawal, please see our Privacy Policy.

Reviewer #1: No

Reviewer #2: No

---

## [Editor Report · Decision Letter 1]

22 Aug 2022

The impact of digital communication and data exchange on primary health service delivery in a small island developing state setting

PDIG-D-22-00057R1

Dear Dr Craig,

We are pleased to inform you that your manuscript 'The impact of digital communication and data exchange on primary health service delivery in a small island developing state setting' has been provisionally accepted for publication in PLOS Digital Health.

Before your manuscript can be formally accepted you will need to complete some formatting changes, which you will receive in a follow-up email from a member of our team. It is also requested that you add a sentence on the methods used in the abstract.

Best regards,

Heather Mattie

Academic Editor

PLOS Digital Health